# Post-RNA (mRNA) Vaccination Myocarditis: CMR Features

**DOI:** 10.3390/diagnostics12051034

**Published:** 2022-04-20

**Authors:** Karuna M. Das, Taleb Al Mansoori, Ali Al Shamisi, Usama MH. AlBastaki, Klaus V. Gorkom, Jamal Aldeen Alkoteesh

**Affiliations:** 1College of Medicine and Health Sciences, United Arab Emirates University, Sheikh Khalifa Bin Zayed Street, Asharej, Al Ain P.O. Box 17666, United Arab Emirates; taleb.almansoor@uaeu.ac.ae (T.A.M.); klausg@uaeu.ac.ae (K.V.G.); 2Tawam Hospital, Al Maqam, Tawam, Abu Dhabi P.O. Box 15258, United Arab Emirates; arshamisi@seha.ae (A.A.S.); jkoteesh@seha.ae (J.A.K.); 3Rashid Hospital, Umm Hurair 2, Dubai P.O. Box 4545, United Arab Emirates; usamauae@gmail.com

**Keywords:** RNA (mRNA) vaccine, CMR, myocarditis

## Abstract

RNA (mRNA) vaccines used to prevent COVID-19 infection may cause myocarditis. We describe a case of acute myocarditis in a 27-year-old male after receiving the second dose of a Pfizer immunization. Three days after receiving the second dose of vaccine, he had acute chest pain. Electrocardiographic examination revealed non-specific ST-T changes in the inferior leads. Troponin levels in his laboratory tests were 733 ng/L. No abnormalities were detected on his echocardiography or coronary angiography. The basal inferoseptal segment was hypokinetic. The LV EF was 50%, whereas the RV EF was 46%. Epicardial and mesocardial LGE were shown in the left ventricle’s basal and mid anterolateral, posterolateral, and inferoseptal segments. The native T1 was 1265 ± 54 ms, and the native T2 was 57 ± 10 ms. Myocardial strain indicated that the baseline values for LV GLS (−14.55), RV GLS (−15.8), and RVCS (−6.88) were considerably lower. The diagnosis of acute myocarditis was determined based on the clinical presentation and cardiac magnetic resonance (CMR) findings.

Myocarditis after vaccination is a rare complication that has received more attention recently following reports of cardiac injury in a limited number of persons after receiving COVID-19 vaccines based on messenger RNA (mRNA) [1,2]. At the end of 2021, about 4.5 billion people worldwide will have received a dose of the COVID-19 vaccine [3]. Thus, despite their rarity, serious adverse events associated with the administration of vaccines targeting COVID-19 are critical for the public, physicians, and other officials. Notably, COVID-19 infection may also result in cardiac damage, which has been associated with poor outcomes in hospitalized patients, and should be evaluated against the risk of vaccine-related complications [3,4]. 

As a result of its unmatched capacity to non-invasively characterize heart tissue, cardiac MRI is critical in evaluating acute myocarditis [5]. Several recent case studies have observed magnetic resonance imaging (MRI) abnormalities in hospitalized patients with myocarditis after COVID-19 immunization. In comparison to other causes of myocarditis, however, the amount of myocardial injury is unknown, especially in non-hospitalized individuals. This report aims to describe the pattern and extent of MRI findings in myocarditis related to COVID-19 immunization and compare these results to those associated with other causes of myocarditis. 

A 27-year-old male presented with malaise and mild fever after a few days of his first COVID-19 Pfizer vaccination. The individual was from an Asian-Arab well-to-do family. He was completely healthy before this episode of illness. After receiving conservative therapy, he improved and received his second vaccination dose. He had severe chest discomfort three days after receiving the second vaccination dose. Initially, the patient was isolated during admission due to the COVID pandemic and a previous history of minor fever and malaise. An RT-PCR test for COVID-19 was performed. Three RT-PCR tests were negative over three days. The electrocardiogram indicated non-specific ST-T alterations in the inferior leads. Laboratory testing indicated that the patient’s electrolytes were normal, and he had a troponin level of 733 ng/L. His echocardiogram and coronary angiography revealed no abnormalities. CMR was performed with a 1.5T scanner (Optima MR450W, GE Healthcare, Waukesha, WI, USA), and the results were consistent with myocarditis. CMR revealed hypokinesia of the basal inferoseptal segment. The LV EF was 50%, while the RV EF was 46%. Epicardial and mesocardial LGE were shown in the LV’s basal and mid-anterolateral, posterolateral, and inferoseptal segments in 2ch and 4ch view (Figure 1A–C). T2w STIR 4ch view revealed increased signal intensity in the anterolateral and lateral segments (Figure 2). The native T1 was 1265 ± 54 ms, and the native T2 was 57 ± 10 ms (Figure 3). Myocardial strain revealed: LV GRS: 23.39, GCS: −15.46, GLS: −14.55 and the RV GRS: 33.28, GCS: −10.74, GLS: −15.8, RVCS: basal: −6.88, mid: −9.95, and apex: −18.92. Based on the modified Lake Louise criteria (LLC), an MRI diagnosis of myocarditis was made [6]. Subsequently, the patient was treated with aspirin, clopidogrel, enoxaparin, fentanyl, and zinc gluconate. Fentanyl was used to relieve severe chest pain within 72 h of admission (50 mcg, intravenously once). He was discharged after a complete recovery.

## Figures and Tables

**Figure 1 diagnostics-12-01034-f001:**
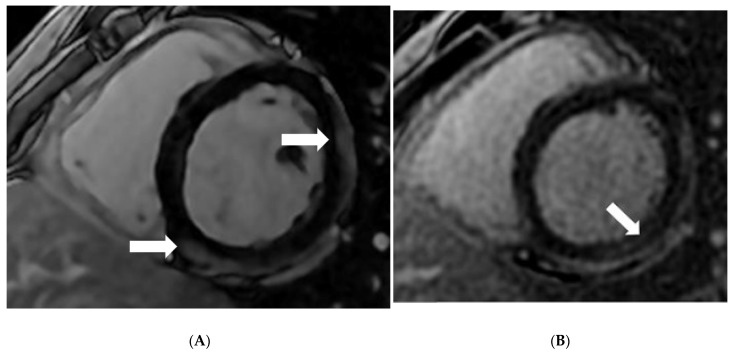
CMR imaging demonstrates epicardial type of late gadolinium enhancement involving the mid-inferoseptal and mid-anterolateral segments (Arrow) (**A**), the midwall type of LGE involving the basal-inferolateral segments (Arrow) (**B**), and multiple patchy type LGE involving the antero-lateral, lateral, with inferior septal midwall (Arrow) (**C**).

**Figure 2 diagnostics-12-01034-f002:**
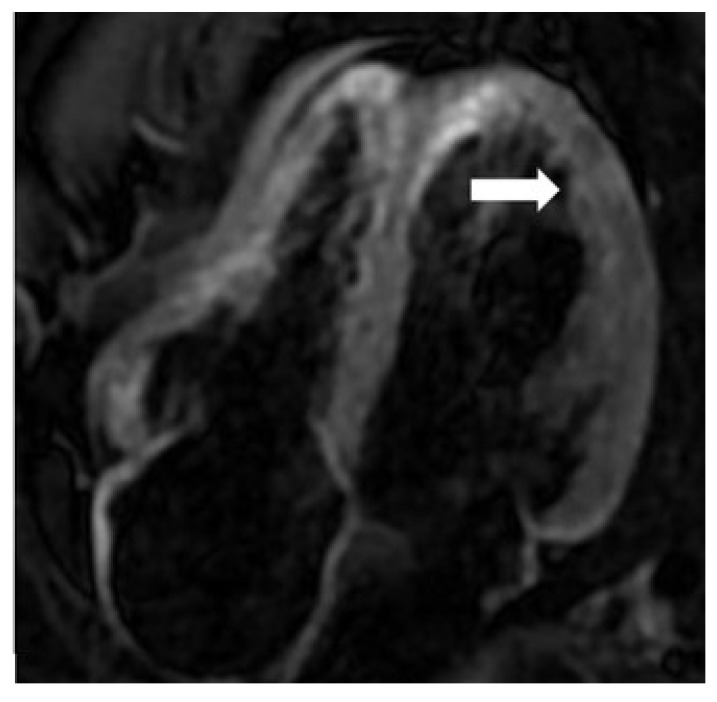
CMR imaging demonstrating T2w STIR 4 ch view with multiple increased signal intensity in anterolateral and lateral segments (arrow).

**Figure 3 diagnostics-12-01034-f003:**
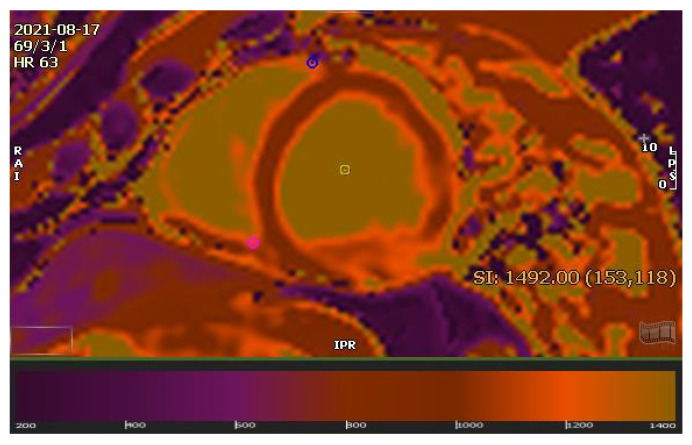
CMR imaging demonstrating T1 mapping. The native T1 is 1265 ± 54 ms. Myocarditis is a well-documented adverse reaction to vaccinations. This study supports the value of CMR in diagnosing myocarditis, which may occur after coronavirus RNA (mRNA) immunizations, and has a favorable prognosis. Additionally, the diagnosis of acute myocarditis seems credible. The patients had acute chest pain and high troponin levels, suggesting myocardial damage. Anomalies in CMR imaging were detected in areas with abnormal wall motion on cine imaging (corresponding regions with LGE and regions with abnormal native T1 and T2 with biventricular reduced ejection fraction). The first patient at our institution with myocarditis following COVID-19 vaccination was observed in line with the timing after the second vaccination dose. Fortunately, the hospital courses of the patients with myocarditis following COVID-19 vaccination were uneventful, and he was discharged within a week. The incidence of myocarditis in the 42 days after the first vaccination dose are 2.13 instances per 100,000 vaccinated people, using established diagnostic criteria [7]. Male patients aged 16 to 29 years had the highest prevalence. The majority of cases were classified as having a mild to moderate degree of severity. Myocarditis was identified throughout the post-vaccination period, with an apparent increase occurring about three to five days after the second vaccine dose. One patient had a cardiogenic shock, while another with a history of cardiac issues died soon after being discharged from the hospital for an unknown cause [7]. Our patient was a 27-year-old male patient who had a biventricular reduced ejection fraction. It was per previous observations where they noted left ventricular dysfunction in 29% of patients with post-vaccine myocarditis [7]. Moreover, our findings are consistent with case studies of hospitalized patients, demonstrating that most patients with vaccine-associated myocarditis are younger men who present symptoms following the second dose, with frequent LGE and myocardial edema on MRI, and rapid improvement in clinical symptoms during short-term follow-up [8,9,10,11]. The modified LLC test was positive in our case, indicating post-vaccine myocarditis, which is consistent with previous experience [12]. The modified LLC’s sensitivity was significantly higher than the sensitivity of the original LLC (*p* = 0.031) [6]. The initial LLC had a sensitivity of 72.5% (95% confidence interval CI: 57.2%, 83.9%) and a specificity of 96.2% (95% CI: 81.1%, 99.3%). The 2018 LLC had a sensitivity of 87.5% (95% CI: 73.9% to 94.5%) and a specificity of 96.2% (95% CI: 81.1%, 99.3%) [6]. The criteria tend to be more accurate during the acute phase of the disease, mainly when an “infarct-like” appearance is present [13]. The length of myocardial inflammation limits the sensitivity of diagnostic imaging to a few weeks following presentation [14]. On the other hand, CMR can provide beneficial characteristics inside this window to help predict prognosis, regardless of clinical symptoms [15,16]. For example, myocardial edema without LGE on CMR has improved recovery and outcomes [16]. Recent developments in T1, ECV, and T2 mapping techniques have enabled quantification of mapping to be included in a modified LLC version [12]. In the acute setting, inflamed myocardium typically exhibits higher T1, T2, and ECV values for extracellular volume. T1 and T2 mapping are estimated to have 89 and 80% diagnostic accuracy, respectively [17], while these statistics are lower in chronic circumstances and vary according to the patient population studied [14]. By contrast, T2 mapping is more selective for inflammation and persists in the subacute period [14]. Additionally, T2 mapping has been demonstrated to aid in the differentiation of active from healed myocarditis [18], with a more substantial negative predictive value for inflammation [19]. A combination method utilizing native T1 mapping and LGE assessment appears to have higher diagnostic accuracy than the LLC criteria in acute myocarditis [5,20]. Our patient’s baseline values for LV GLS (−14.55), RV GLS (−15.8), and RVCS were all considerably lower (−6.88). CMR-FT is lowered in patients with suspected myocarditis in acute myocarditis [21]. A recent study showed that ventricular CMR-FT progressively enhanced diagnostic performance for myocarditis compared to traditional LLC, particularly in patients with maintained ejection fraction. Our CMR imaging data also show that myocarditis related to COVID-19 immunization is typically mild to moderate in severity, similar to myocarditis associated with other etiologies, and those previously described [20,22]. CMR imaging indicated favorable results in most patients: normal or slightly reduced left ventricular ejection fraction, and a small amount of LGE. Interestingly, the findings corroborate those of Puntmann et al. [23]. A manifestation of myocarditis associated with COVID-19 RNA (mRNA) immunization was documented. Clinical characteristics, CMR imaging findings, and short-term clinical outcomes indicate a favorable clinical outcome. Even though patient demographics vary and MRI findings are usually milder, our analysis demonstrates that the pattern of MRI abnormalities in vaccine-associated myocarditis is akin to that reported in other cases.

## Data Availability

Not applicable.

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
