# Peer review of "Post-RNA (mRNA) Vaccination Myocarditis: CMR Features"

_diagnostics, 2022, doi:10.3390/diagnostics12051034_

Round 1

Reviewer 1 Report

The paper is well organized and very useful for the effects of the RNA vaccine. I would just like to report that the therapy administered to the patient should be perfected as follows: why the choice of this therapy, the indications that these drugs have (for example fentalyl for pain? Or for other reasons?), Duration and how many times a day for each of them. they and whether the patient benefited from them.The paper is well organized and very useful for the effects of the RNA vaccine. I would just like to report that the therapy administered to the patient should be perfected as follows: why the choice of this therapy, the indications that these drugs have (for example fentalyl for pain? Or for other reasons?), how many days, and how many times a day for each of them. they and whether the patient benefited from them.

Author Response

I would like to thank the honorable reviewer for the comment. It was given only once at the initial admission (first 72 hours) (fentanyl (N): 50 mcg, 1 mL, IV, Once.) because of severe chest pain, and he got perfect relief of the pain. Subsequently, he was discharged after five days, and the same drug was no longer used. 

Reviewer 2 Report

I commend the authors for the work that adds important evidence regarding a complication of the anti-Covid-19 vaccination. In fact, it is now known that the rate of myocarditis that occurs following anti-covid vaccination is higher than that of the general population.

The authors report a clinical case in the form of a case report detailing the essential elements of the case in a concise and clear way and placing emphasis on the sensitivity of instrumental diagnosis through cardiac magnetic resonance.

The theme is interesting and current, in fact, any evidence that manages to shed light on a complication of vaccination will in the future help to diagnose and treat these complications early by placing patients in a specific path.

Before publication I suggest some changes and I would like to have some clarifications from the authors.

  1. Although the use of abbreviations is in common use, I still consider it appropriate in drafting the article to use the word in full the first time and then use the abbreviation and not vice versa. For example, in the Introduction to the second paragraph, "MRI" is read for the first time and then "magnetic resonance imaging" is written in full. I suggest the authors to reverse the wording and first use "magnetic resonance imaging (MRI)" and then only the abbreviation.
    Also for CMR, also present in the title, we could proceed in the same way.

2.2. As regards the "case report" chapter, I ask if it is possible to have further information regarding the patient concerned. Such information can be both socio-demographic and clinical.
For example, among the useful information I think of previous COVID-19 infection.
An interesting work published in Circulation (https://doi.org/10.1161/CIRCULATIONAHA.121.056583Circulation. 2022; 145: 345–356) reports a large number of cases of suspected myocarditis and an important role is also given to imaging ( lake louise criteria). This work also emphasizes the fact that myocarditis is also an event linked to the natural history of the disease. I advise the authors to deepen this work which, I believe, deserves to be cited in order to update the bibliography as much as possible.

Author Response

We want to thank the honorable reviewer for his valuable comment. As per his suggestion, we have elaborated on the original name preceding the abbreviation. The correction is made in the title and the text. 

The individual was from an Asian-Arab well-to-do family. He was completely healthy before this episode of illness. Initially, the patient was isolated due to the Covid pandemic and a minor fever and malaise. An RT-PCR test for Covid-19 was performed. Three tests were negative over three days.  We have reviewed the article suggested by the honorable reviewer and have incorporated it in the bibliography. Apart from this, we have also highlighted the role of LLC with the addition of another 11 references about the present status of LLC and modified LLC in the diagnosis of myocarditis.